# Inclusion of Sunflower Oil, Organic Selenium, and Vitamin E on Milk Production and Composition, and Blood Parameters of Lactating Cows

**DOI:** 10.3390/ani12151968

**Published:** 2022-08-03

**Authors:** Arlindo Saran Netto, Thiago H. Silva, Mellory M. Martins, Ana Maria C. Vidal, Márcia S. V. Salles, Luiz C. Roma Júnior, Marcus A. Zanetti

**Affiliations:** 1Department of Animal Science, College of Animal Science and Food Engineering, University of São Paulo, Pirassununga 13635-900, SP, Brazil; silvath@usp.br (T.H.S.); mellory.martins@usp.br (M.M.M.); mzanetti@usp.br (M.A.Z.); 2Department of Veterinary Medicine, College of Animal Science and Food Engineering, University of São Paulo, Pirassununga 13635-900, SP, Brazil; anavidal@usp.br; 3São Paulo Agency for Agribusiness Technology (APTA), Ribeirão Preto 14030-680, SP, Brazil; marcia.saladini@gmail.com (M.S.V.S.); lcromajr@gmail.com (L.C.R.J.)

**Keywords:** antioxidants, biofortified, functional food, milk quality

## Abstract

**Simple Summary:**

Feeding sunflower oil, selenium, and vitamin E to lactating dairy cows has improved the nutritional profile of milk for human consumption and positively impacted animal performance. This may be attributed to the increased healthier fat components, i.e., “good fats”, and antioxidant substances in milk. This study evaluated the effects of supplementing sunflower oil, selenium, and vitamin E on milk production and composition, and the blood parameters of lactating dairy cows. Supplementing sunflower oil to lactating dairy cows provided beneficial effects on milk fatty acid profiles, increasing healthier fatty acids concentrations, which have been reported as important anticarcinogenic, antiatherogenic, and antidiabetic nutrients in human diet. However, this strategy reduced the milk fat content. Selenium and vitamin E supplementation improved milk production and provided higher selenium and vitamin E content in blood and milk. These compounds are important antioxidants and nutrients for animal and human health.

**Abstract:**

Aiming to improve milk quality and animal health, the effects of the inclusion of sunflower oil with added organic selenium (Se) and vitamin E in the diets of lactating cows were evaluated. Twenty-four multiparous lactating Jersey cows were randomly enrolled into four treatments: CON (control); SEL [2.5 mg organic Se kg^−1^ dry matter (DM) + 1000 IU vitamin E daily]; SUN (sunflower oil 3% DM); and SEL + SUN (sunflower oil 3% DM + 2.5 mg organic Se kg^−1^ DM + 1000 IU vitamin E daily). The experimental period was 12 weeks with 14 days for acclimation. Cows were milked twice a day. Dry matter intake, milk production, and composition were measured daily and analyzed in a pooled 4-week sample. On day 84, white blood cell counts, as well as serum and milk Se and vitamin E levels, were assessed. Supplementation with selenium and vitamin E alone or combined with sunflower oil increased milk production, and increased the serum and milk concentrations of those nutrients. The inclusion of sunflower oil reduced fat content and DM intake but also altered the milk fatty acid profile, mainly increasing levels of *trans* 11 C18:1 (vaccenic) and *cis* 9 *trans* 11 conjugated linoleic acid (CLA). Our results indicate that supplementation with sunflower oil, Se and vitamin E provides beneficial effects on animal performance and milk composition, which could be an important source of CLA and antioxidants (Se and vitamin E) for human consumption.

## 1. Introduction

Dietary inclusion of lipid sources and antioxidants for lactating cows has been suggested as an alternative means by which to alter the milk fatty acid profile and increase milk production, respectively [1,2]. Oliveira et al. [3] reported that dietary inclusion of polyunsaturated fatty acids for dairy cows altered the milk fatty acid composition. Thus, sunflower oil, a rich source of polyunsaturated fatty acids (PUFA’s), could modulate the ruminal biohydrogenation process, which decreases saturated fatty acids and increases polyunsaturated fatty acid levels in milk [4]. Staples et al. [5] reported that the inclusion of fatty acids in the diet of dairy cows might be an interesting alternative for improving energy input to animals during challenging physiological periods, such as the transition period and the peak milk yield period, i.e., around 4–6 week after calving. In addition, unsaturated fatty acid supplementation results in healthier milk for human consumption. Notably, higher omega-3 and conjugated linoleic acid (CLA) concentrations have been detected in such milk, which give rise to lower atherogenicity and thrombogenicity indexes [6].

Dietary supplementation of selenium (Se) and vitamin E increases the levels of these antioxidants in milk [7,8]. Lauzon et al. [9] reported that antioxidants may be effective in protecting the mammary glands of dairy cows against oxidative stress. Also, Yang et al. [10] observed that Se and vitamin E are important antioxidants which protect cellular membranes from reactive oxygen species and lipid hydroperoxides. Additionally, Se and vitamin E contribute to growth, reproduction, the immune system, hormone synthesis, and tissue integrity, and have also been related to reducing somatic cell count and increasing productivity and milk quality [1,11,12,13,14,15,16]. Since organic sources of selenium may alter ruminal fermentability [17], protecting the microbiota against membrane cell oxidative damage [18], this mineral coupled to vitamin E might mitigate the negative impact of free-oils supplementation on rumen performance, maintaining high levels of productivity and improving the milk quality. However, to date, only one study has reported the effect of dietary sunflower oil and Se plus vitamin E together in lactating dairy cows [2].

Our hypothesis was that the inclusion of sunflower oil and Se plus vitamin E would enhance milk production and its composition, increasing the proportion of unsaturated (healthier) fatty acids along with increasing antioxidant concentration. 

## 2. Materials and Methods

### 2.1. Animals and Diets

All experimental procedures were approved by the University of São Paulo, College of Animal Science and Food Engineering (USP/FZEA), Ethics Committee for the Use of Animals in Experiments (#012606). This study was conducted at the São Paulo Agency for Agribusiness Technology [Agência Paulista de Tecnologia dos Agronegócios (APTA)], in Ribeirão Preto, Brazil, from August to December 2009. This location (47°51′ W and 21°12′ S; altitude 646 m) is considered tropical humid, with average rain precipitation of 1427 mm, an average maximum temperature of 25 °C, and an average minimum temperature of 19 °C. 

A total of 24 multiparous lactating Jersey cows (2.4 ± 0.5 lactations; 370.8 ± 35.7 kg of body weight; 62 ± 10 days in milk; and initial milk yield of 12.1 ± 1.8 kg/d, mean ± SD), were used in a completely randomized design trial. The cows were randomly assigned into one of four treatments according to the Microsoft Excel RAND command: (1) Control (CON) cows were fed a basal ration without sunflower oil, an organic mineral source or additional vitamins; (2) Selenium (SEL) cows were fed the basal ration with added organic Se (selenium yeast; Sel-Plex^®^, Alltech Inc., Nicholasville, KY, USA) at 2.5 mg kg^−1^ dry matter (DM), and 1000 IU of vitamin E, daily; (3) Sunflower oil (SUN) cows received the basal ration with 3% sunflower oil kg^−1^ DM; and (4) Selenium + sunflower oil (SEL + SUN) cows were fed the basal ration with added organic Se (selenium yeast; Sel-Plex^®^, Alltech Inc.) at 2.5 mg kg^−1^ DM, 1000 IU of vitamin E, and 3% of sunflower oil kg^−1^ DM (Table 1). The total mixed ration (TMR) had 50% of roughage (corn silage) and 50% concentrate (DM-basis), according to recommendations described in [19]. The sunflower oil, Se, and vitamin E were mixed in TMR. The cows were housed in individual pens (17.5 m^2^), with sand bedding, feed bunks, and forced ventilation. The adaptation period to the diet was 14 days.

The experimental TMR was provided for 12 weeks after the adaptation period in equal amounts, twice daily (0700 h and 1400 h). Feed intake was determined based on daily feed offered and refusals, in which the latter was kept to 50–100 g/kg of total offered feed. Feed and refusals were collected weekly and stored at −20 °C. The dry matter intake (DMI) and chemical analyses of the experimental diets were performed for each cow in a pooled 4-week sample (weeks 3, 6, 9, and 12). All samples were assessed for dry matter (DM, AOAC 950.15), total nitrogen (AOAC, 984.13) for crude protein (CP) determination (6.25 × total nitrogen), ether extract (EE, AOAC 920.39), and ash (AOAC 942.05), according to the method described in [20]. The neutral detergent fiber (aNDF) and acid detergent fiber (aADF) content of samples were determined as stated by [21]. An aNDFanalysis was performed using amylase without sodium sulfide (TE-149 fiber analyzer, Tecnal Equipment for Laboratory Inc., Piracicaba, Brazil). Dry matter intake was quantified as the mean over the entire experimental period.

**Table 1 animals-12-01968-t001:** Ingredients and chemical composition of the experimental diets (values expressed as % of DM, otherwise stated) ^1^.

Item	Treatments ^1^			
	CON	SEL	SUN	SEL + SUN
Ingredients				
Corn silage	50.0	50.0	50.0	50.0
Corn meal	25.0	25.0	22.0	22.0
Soybean meal	18.0	18.0	18.0	18.0
Wheat bran	4.0	4.0	4.0	4.0
Urea	0.9	0.9	0.9	0.9
Salt	0.5	0.49	0.5	0.49
Mineral premix ^2^	1.0	1.0	1.0	1.0
Ammonia sulfate	0.04	0.04	0.04	0.04
Sodium bicarbonate	0.53	0.53	0.53	0.53
Sunflower oil ^3^	-	-	3	3
Selenium, mg/kg	-	2.5	-	2.5
Vitamin E, IU	-	1000	-	1000
Chemical composition				
DM	57.7	57.8	58.3	58.2
Ash	6.4	6.4	6.5	6.6
CP ^4^	18.1	18.6	18.4	18.4
Ether extract	2.3	2.4	3.6	3.4
Nitrogen-free extract	57.5	57.0	56.3	56.2
aADF	21.1	21.3	21.6	21.3
aNDF	37.9	37.8	37.7	37.1
NE_L_, Mcal/kg DM ^5^	1.70	1.69	1.82	1.81

^1^ CON = control total mixed ration; SEL = total mixed ration formulated with 2.5 mg Se kg^−1^ dry matter (DM) and 1000 IU vitamin E; SUN = total mixed ration formulated with 3% sunflower oil; SEL = total mixed ration formulated with 3% sunflower oil, 2.5 mg Se kg−1 DM and 1000 IU vitamin E. ^2^ Mineral premix (composition per kg): 80 g S, 20 g Mg, 20 g K, 1000 mg Mn, 2500 mg Zn, 1500 mg Cu, 100 mg Co, 80 mg I, 20 mg Se, 180 g Ca, 90 g P, 300 mg F (max.). ^3^ Sunflower oil fatty acid profile (as % of total fatty acids): 6:0, 8:0, and 10:0—not detected; 12:0—up to 0.1; 14:0—up to 0.2; 16:0—5 to 7.6; 16:1—up to 0.3; 17:0—up to 0.2; 17:1—up to 0.1; 18:1—14 to 39.4; 18:2—48.3 to 74; 18:3—up to 0.3; 20:0—0.1 to 0.5; 20:1—up to 0.3; 20:2—not detected; 22:0—0.3 to 1.5; 22:1—up to 0.3; 22:2—up to 0.3; 24:0—up to 0.5; and 24:1—not detected. ^4^ Crude protein. ^5^ Net energy of lactation was calculated according to Weiss et al. [22]: NEL (Mcal/kg DM) = 0.024 × TDN (%) − 0.12, using 1 as the processing factor.

### 2.2. Milk Production and Composition

Cows were milked twice a day (0600 h and 1600 h), and milk yield was recorded using an electronic device (Alpro^®^, DeLaval, Tumba, Sweden). Pooled milk samples for each cow, from two consecutive milkings and keeping a fixed ratio, were collected weekly. All milk analyses were performed in a pooled sample from four experimental weeks (weeks 3, 6, 9, and 12), including milk yield information. Fat-corrected milk (FCM, 3.5%) was calculated according to the method proposed by Sklan et al. [23], wherein FCM = (0.432 + 0.1625 × percentage of fat) × kg of milk. Plastic bottles containing preservative potassium dichromate tablets were used for milk storage and transport to the laboratory. Milk chemical components were determined by infrared (Bentley 2000^®^, Bentley Instruments Inc., Chaska, MN, USA), and SCC by flow cytometry (Somacount 300^®^, Bentley Instruments Inc., Chaska, MN, USA). Milk vitamin E concentration was analyzed using high-performance liquid chromatography (HPLC) at the Institute of Food Technology [Instituto de Tecnologia de Alimentos (ITAL)] in Campinas, Brazil, according to [24]. Briefly, 1 g of whey of milk was weighed into a test tube with 7.3 mL of the saponification solution (11% *v*/*v* KOH, 45% *v*/*v* H_2_O, 55% *v*/*v* ETOH, and 0.25 g vitamin C/sample). Four milliliters of isooctane were added to each sample and the tubes were vortexed for 2 min to extract the vitamin E. The tubes were rested to separate the isooctane from the water, which was transferred to a vial for HPLC analysis. Milk Se concentration was analyzed using a wet digestion mixture with nitric-perchloric acid, and a fluorometric reading was subsequently taken, followed by diaminonaphthalene sensitization [25] at the USP/FZEA Mineral Laboratory in Pirassununga, Brazil.

The milk relative density was determined using a hydrometer at 15 °C. The milk fat content was estimated by the Gerber method. The total dry extract (TDE) was estimated by gravimetry. The non-fat dry extract (NDE) was estimated by the following equation: NDE = %TDE − %fat. These analyses were carried out at the Clínica do Leite Laboratory, University of São Paulo, School of Agriculture Luiz de Queiroz (USP/ESALQ). 

For milk fatty acid profile determination, fat extraction was performed by the method described in [26]. Then, the separated fat (300–400 mg) was methylated and sterols were formed according to the method described in [27], and fatty acids were quantified by gas chromatography (GC-2010 Plus, with on-column automatized injection, Shiamdzu Co., Kyoto, Japan) using a capillary column (100 m × 25 mm diameter, 0.02 mm thick; Suppelco, Bellfonte, PA, USA). Hydrogen was used as carrier gas at a flow rate of 40 cm/s, and the vaporizer and detector temperatures were 200 and 300 °C, respectively. The oven temperature increased from 70 °C to 230 °C at rates of 13, 4, and 7 °C/min for the three different phases of temperature raising. During the identification procedure, four standards were utilized: C4-C24 standard fatty acid (Supelco^®^ 37 Component FAME Mix, Sigma-Aldrich Co, St. Louis, MO, USA); C18:1 *cis*-vaccenic acid (V0384-1G SIGMA, Sigma-Aldrich Co, St. Louis, MO, USA); C18:2 10-*trans*, 12-*cis* methyl (UC-61-M, Nu-Check Prep, Inc., Elysian, MN, USA); and C18:2 9-*cis*, 11-*trans* methyl (UC-60-M, Nu-Check Prep, Inc., Elysian, MN, USA).

### 2.3. Blood Sampling and Analysis

On day 84, blood samples were collected by puncture of the coccygeal vessels. Samples were placed in vacutainer tubes without anticoagulant but with potassium EDTA (Becton, Dickinson and Company, Franklin Lakes, NJ, USA). The coagulated and EDTA were analyzed or processed within 1 h after collection. The blood samples from tubes without anticoagulant were centrifuged at 3000× *g* for 15 min at 4 °C for serum separation and frozen at −80 °C for further analysis. White cell count was performed using a Neubauer chamber, utilizing a Thoma pipette after dilution to 1:20, using Türk solution. Serum concentration of vitamin E was determined according to the method described in [24] using liquid chromatography. Serum Se concentration was assessed using a fluorometric method, as described by Olson et al. [25]. Vitamin E analysis was performed at the Diagnostics and Clinical Analysis Laboratory in Pirassununga, and Se analysis was performed at the USP/FZEA Mineral Laboratory.

Blood serum was analyzed for glucose, total cholesterol, high- (HDL), low- (LDL), and very low-density lipoprotein (VLDL) cholesterol, and triglycerides using commercial kits (Laborlab^®^; Guarulhos, SP, Brazil) by an enzymatic-colorimetric method. 

### 2.4. Statistical Analysis

All statistical analyses were performed in SAS version 9.4 SAS (SAS Institute Inc., Cary, NC, USA). All variables were analyzed using the MIXED procedure according to the following model:$$Y_{\mathrm{ij}}=µ+\alpha_{i}+\beta (\mathrm{DMI}-\bar{\mathrm{DMI}})+A_{j}+e_{\mathrm{ij}}$$
where Y_ij_ = represents the observation for animal j in a given treatment i; µ = is the intercept; α_i_ = fixed effect of ith treatment (i = 1…4); β = linear regression coefficient of initial DMI of each cow; A_j_ = random effect of the jth animal (j = 1…24) ≈ N (0;σA^2^); e_ij_ = random error associated with each observation ≈ N (0;σ_e_^2^); N = Gaussian distribution; σA^2^ = estimated variance associated with animals; and σ_e_^2^ = estimated residual variance. Visual assessment of the distribution plots of the studentized residuals was used to confirm the normality of the distribution. As the values of SCC and white blood cell count did not present a normal distribution, they were transformed to log_e_. Means were compared using a Bonferroni adjustment option. Means were obtained by Least Squares Means (LSMEANS). For all analyses, differences detected at *p* < 0.05 were considered significant.

## 3. Results and Discussion

### 3.1. Chemical Composition of Diets

As shown in Table 1, the ether extract content was numerically higher in the SUN and SEL + SUN groups compared to the CON and SEL groups (2.3, 2.4, 3.6, and 3.4% EE in CON, SEL, SUN, and SEL + SUN treatments, respectively). However, CP, aADF and aNDF were similar among treatments. These descriptive results confirm the isolated effect of EE and antioxidant inclusion on the results depicted herein. 

### 3.2. Performance and Milk Composition

Less DM intake was observed in cows fed with sunflower oil (*p* < 0.05; Table 2). The mechanism by which this supplementation influenced intake has not been fully elucidated. Strong evidence has suggested that the reduced intake is related to the effects of fat on ruminal fermentation, motility, poor acceptability, intestinal hormones release, regulatory mechanisms controlling feed intake, and the restrict ability of ruminants to oxidize fatty acids [28]. Also, DMI reduction may be related to plasma concentrations of certain fatty acids resulting from fat metabolism. The fatty acids that appear to be involved in the DMI reduction mechanism are C18:2 n-6 and C18:1 n-9 [29], which represent, respectively, 63.42 and 23.64% of the fatty acid profile of sunflower oil [30].

The use of antioxidants in diets resulted in greater milk yield (*p* = 0.03; Table 2). The nutrition program of a dairy herd has great influence on cow productivity and health [31]. The use of organic Se sources and vitamin E provides better immune response, greater resistance to udder and mammary infections, and, consequently, enhancement of milk production due to their role in antioxidant systems [1]. 

Differences were also detected for total milk fat content among treatments (*p* = 0.04). Treatments with oil inclusion (SUN and SEL + SUN) had less fat content compared to those without sunflower oil inclusion (CON and SEL; Table 2). Some studies [32,33,34,35] have explained the changes in content and composition of milk fat by the traditional glycogen/insulin theory, i.e., an increase of starch intake enhances concentrations of circulating insulin, increases deposition of adipocytes, and decreases acetate for milk fat synthesis in the mammary gland. However, the most acceptable theory was reported in [36]. The authors of that paper described that the inhibition of milk fat synthesis, as well as its modified fatty acid composition, may be related to substances produced in the rumen such as the polyunsaturated fatty acids. Some studies have shown that the isomer *trans*-10, *cis*-12 CLA, produced during the intense process of ruminal biohydrogenation of free-oil supplemented dairy cows, inhibits milk fat synthesis in the mammary gland [37,38,39].

Higher milk Se content was detected for Se supplemented cows (*p* < 0.01; Table 2). Regarding vitamin E, the inclusion of sunflower oil, associated or not with Se and vitamin E, increased vitamin E levels in milk (29.3% for SEL, 92.7% for SUN, and 263.4% for SEL + SUN, compared to CON; *p* < 0.05). Weiss and Wyatt [40], evaluating different levels of vitamin E and fat sources, reported an increase in plasma and milk α-tocopherol levels which was accentuated in dairy cows fed with fat, as reported in the current study. The absorption of this liposoluble vitamin is linked to fat digestion, which depends on micellar solubilization and is facilitated by bile and pancreatic lipase [41]. Lynch et al. [42] also reported that vitamin E supplementation increased α-tocopherol levels in beef. Due to their importance to human health, Se- and vitamin E-biofortified milk could be an essential and low-cost source of these nutrients for human consumption, with a positive effect on lipid metabolism. Insufficiency of these nutrients is a global concern due to problems related to cardiovascular disease [43]. 

The milk chemical composition and references according to the Brazilian Regulation of Industrial and Sanitary Inspection of Animal Products (RIISPOA; [44]) are summarized in Table 3. Analyses of milk chemical composition are important, because milk is considered one of the most complete foods (in terms of nutritional value) for human health. It can also be an excellent substrate for a wide variety of microorganisms, including pathogens. The authors of [44] noted that to be considered eligible for human consumption, milk must have physicochemical properties according to the reference values. In our study, all diets resulted in the production of milk with excellent quality for human consumption, meeting all the Brazilian legislation requirements.

### 3.3. Milk Fatty Acids Profile

Sunflower oil with or without antioxidants reduced the proportion of short- and medium-chain fatty acids, as reflected in lower <C16 and saturated fatty acid contents in these treatments (*p* < 0.05, *p* = 0.03, and *p* = 0.01, respectively; Table 4). This reduction was expected, because supplementation with vegetable oils rich in long-chain fatty acids inhibits the enzymatic complex involved in de novo synthesis and reduces acetate and beta-hydroxybutyrate [45]. Cows fed with Se (SEL), sunflower oil (SUN), and their combinations had less (*p* = 0.01) milk C15; however, there was no effect on milk C17. Conversely, the proportion of C18:1 trans-11 (vaccenic) and CLA *cis*-9, trans-11 increased (*p* < 0.001). Since trans- are less toxic than *cis*-fatty acids, the increased production of trans is a defense mechanism of the microbiota, ensuring the integrity of the cell membrane. Increased vaccenic concentration is important because it is a precursor of endogenous *cis*-9, *trans*-11 CLA, which is beneficial to animals and human health due to its ability to prevent different types of cancer, hypertension, atherosclerosis, and diabetes, and to improve immune system function [46]. In this study, sunflower oil plus antioxidant inclusion led to a different milk fatty acids profile compared to the pure sunflower oil supplementation. It is known that highly digestible organic sources of selenium, such as that applied herein, may alter ruminal fermentability [17]. A plausible explanation is that ruminal microbes incorporate Se to form their protein and cell wall components, which protect them against membrane cell oxidative damage. This phenomenon may modify the ruminal microbiota composition, modulating the ruminal fermentation pattern [18].

### 3.4. White Blood Cell and Serum Antioxidants Levels

No effect was detected on white blood cell counts among treatments (*p* < 0.0001; Table 5). Cows supplemented with Se and sunflower oil had greater serum levels of vitamin E (*p* = 0.03; Table 5), especially SEL + SUN cows, which had 33.5% greater vitamin E concentrations compared to CON cows. This effect may be due to the aforementioned physiological process that associates higher liposoluble vitamin absorption and fat digestion [41]. In addition, the inclusion of dietary Se increased its concentration in serum, compared to treatments without Se (*p* < 0.001; Table 5). Likewise, Weiss et al. [13] reported that plasma Se concentrations of dairy cows were correlated positively with intakes of Se below 5 mg/d, but were independent of Se intake above 5 mg/d. Furthermore, the authors reported that vitamin E intake was positively correlated to plasma vitamin E concentrations, but that vitamin E intake had a greater effect on serum vitamin E values in dry cows than in lactating cows. Corroborating this, Zanetti et al. [47] reported that oral supplementation with 5 mg Se increased serum levels of this mineral in Holstein lactating cows, reducing the incidence of subclinical mastitis. However, herein, no effect of the treatments was detected on SCC or on white blood cell count. 

### 3.5. Serum Metabolites

The treatments had no effect on serum levels of cholesterol, HDL, triglycerides, or glucose (*p* > 0.10; Table 6). However, the inclusion of sunflower oil, Se, and vitamin E decreased serum levels of LDL and VLDL compared to CON cows (*p* < 0.05; Table 6). Excessive dietary fatty acids, after conversion into triacylglycerol in the liver, are transported via VLDL to muscles and adipose tissue. Then, after losing some triglycerides, VLDL becomes LDL, which transports cholesterol into peripheral tissues. This demonstrates that there is a direct relation between VLDL and LDL, corroborating our results. Regarding their reduction, different combinations of lipids and proteins may produce particles with different densities such as HDL, LDL, and VLDL [48]. Furthermore, dietary Se may reduce VLDL- and LDL-cholesterol, regulating the HMG-CoA reductase enzyme by decreasing glutathione peroxidase (GSH) and increasing oxidized glutathione (GSSG) activities [49].

## 4. Conclusions

Supplementation with sunflower oil and its combination with antioxidants positively altered the DMI, milk yield, and fat-corrected (3.5%) milk yield of lactating Jersey cows. Moreover, our results demonstrate that dietary inclusion of sunflower oil increased healthier fatty acid concentrations in milk and reduced low- and very low-density cholesterol (bad cholesterols) in blood. Furthermore, selenium and vitamin E supplementation increased these nutrients in blood and, hence, in milk. This study demonstrates the positive effect of Se plus vitamin E supplementation on mitigating the negative effect of free-oil supplementation to dairy cows. Further, feeding animals with sunflower oils and antioxidants (selenium and vitamin E) may be an effective way to biofortify milk, which might positively impact milk consumers’ health.

## Figures and Tables

**Table 2 animals-12-01968-t002:** Feed intake, milk yield, and composition of lactating Jersey cows fed supplements with selenium, vitamin E and sunflower oil.

Item		Treatments ^1^				
		CON	SEL	SUN	SEL + SUN	SEM ^2^
DMI, kg/d		12.5 ^a^	12.6 ^a^	11.0 ^c^	11.7 ^b^	0.40
Milk yield, kg/d		13.4 ^b^	17.2 ^a^	14.0 ^b^	16.6 ^a^	0.60
Fat-corrected milk yield (3.5%), kg/d		14.1 ^c^	17.8 ^a^	13.6 ^c^	15.4 ^b^	0.45
Milk efficiency, 3.5% MP/kg of DMI		1.1	1.4	1.2	1.3	0.17
Fat, %		3.8 ^a^	3.7 ^a^	3.3 ^b^	3.1 ^b^	0.12
Protein, %		3.5	3.4	3.3	3.3	0.02
Lactose, %		4.5	4.5	4.5	4.7	0.03
NDE ^3^, %		9.1	8.9	8.8	8.9	0.45
Total dry extract, %		12.5	12.6	12.1	12.0	0.40
SCC ^4^, log_e_		4.85	4.54	5.61	4.42	0.38
Selenium, µg/mL		0.042 ^b^	0.09 ^a^	0.04 ^b^	0.10 ^a^	0.01
Vitamin E, mg/mL		0.41 ^d^	0.53 ^c^	0.79 ^b^	1.49 ^a^	0.02
Milk component yield, kg/d						
	Fat	0.53	0.65	0.48	0.52	0.10
	Protein	0.47	0.58	0.46	0.54	0.02
	Lactose	0.61	0.78	0.62	0.79	0.02

^a–d^ Means within a row with different superscripts differ by Bonferroni test (*p* < 0.05). ^1^ Cows were provided four experimental diets: CON = control total mixed ration; SEL = total mixed ration formulated with 2.5 mg Se kg^−1^ DM and 1000 IU vitamin E; SUN = total mixed ration formulated with 3% sunflower oil; SEL = total mixed ration formulated with 3% sunflower oil, 2.5 mg Se kg^−1^ DM and 1000 IU vitamin E. ^2^ Pooled standard error (SEM). ^3^ Non-fat dry extract. ^4^ Somatic cell count.

**Table 3 animals-12-01968-t003:** Average values of milk chemical composition and references according to the Brazilian Regulation of Industrial and Sanitary Inspection of Animal Products.

Item	Values	Reference Values ^1^
Relative density at 15 °C, g/cm^3^	1.032	1.028–1.034
Titratable acidity, Dornic	18	14–18
Fat, %	3.4825	minimum 3
Total dry extract, %	12.3	minimum 11.4
Non-fat dry extract, %	8.9075	minimum 8.4

^1^ References from the Brazilian Regulation of Industrial and Sanitary Inspection of Animal Products [Regulamento da Inspeção Industrial e Sanitária de Produtos de Origem Animal (RIISPOA); [44]].

**Table 4 animals-12-01968-t004:** Effect of experimental diets on milk fatty acid profile composition.

	Treatments ^1^				
Item	CON	SEL	SUN	SEL + SUN	SEM ^2^
C4	1.01 ^a^	1.05 ^a^	0.89 ^b^	0.65 ^c^	0.213
C6	1.15 ^a^	1.18 ^a^	1.03 ^b^	0.93 ^b^	0.184
C8	0.85 ^a^	0.84 ^a^	0.75 ^b^	0.68 ^b^	0.143
C10	2.32	2.24	2.05	1.90	0.440
C12	2.95	2.75	2.59	2.29	0.591
C14	10.38 ^a^	9.65 ^a^	8.47 ^a^	8.11 ^b^	1.634
C15	0.93 ^a^	0.80 ^b^	0.85 ^b^	0.74 ^b^	0.143
C16	30.13	27.65	28.30	26.23	3.742
C16:1 *cis*	0.98	0.65	0.87	0.83	0.302
C17	0.37	0.32	0.33	0.33	0.051
C18	7.50	9.09	7.53	8.10	1.600
C18:2 *cis*	0.88	0.96	0.89	0.93	0.154
C18:1 *trans*-11	0.14 ^b^	0.12 ^b^	0.20 ^a^	0.15 ^b^	0.053
C18:1 *cis*-9	12.67	11.72	11.15	13.37	3.082
C20	0.08	0.08	0.05	0.08	0.033
CLA *cis*-9, *trans*-11	0.13 ^b^	0.07 ^b^	0.19 ^a^	0.10 ^b^	0.063
Total					
Saturated	57.9 ^a^	55.8 ^a^	53.0 ^ab^	50.2 ^b^	6.871
Unsaturated	8.36	7.65	8.14	7.92	0.622
Unsat.:Sat. ratio	0.14	0.14	0.15	0.16	0.744
Total					
Sat. C:18	7.50	9.09	7.53	8.10	1.600
Unsat. C:18	6.01	6.07	6.08	6.06	0.182
Uns:Sat C:18	1.24	1.50	1.24	1.34	0.263
<C16	21.1 ^a^	19.6 ^a^	17.9 ^a^	16.4 ^b^	3.333
>C16	14.27	15.79	14.31	14.77	1.722
C16	31.11	28.31	29.18	27.06	3.864

^a–c^ Means within a row with different superscripts differ by Bonferroni-test (*p* < 0.05). ^1^ Cows were provided four experimental diets: CON = control total mixed ration; SEL = total mixed ration formulated with 2.5 mg Se kg^−1^ DM and 1000 IU vitamin E; SUN = total mixed ration formulated with 3% sunflower oil; SEL = total mixed ration formulated with 3% sunflower oil, 2.5 mg Se kg^−1^ DM and 1000 IU vitamin E. ^2^ Least square means with pooled standard error (SEM).

**Table 5 animals-12-01968-t005:** White blood cell counts and serum concentrations of selenium and vitamin E of lactating Jersey cows fed supplements containing selenium, vitamin E and sunflower oil.

Item	Treatments ^1^				
	CON	SEL	SUN	SEL + SUN	SEM ^2^
White blood cell, log_e_	9.45	9.28	9.46	9.20	0.324
Vitamin E, µg/mL	4.93 ^c^	5.63 ^b^	4.24 ^c^	6.58 ^a^	0.121
Selenium, µg/mL	0.065 ^b^	0.12 ^a^	0.07 ^b^	0.14 ^a^	0.020

^a–c^ Means within a row with different superscripts differ by Bonferroni-test (*p* < 0.05). ^1^ Cows were provided four experimental diets: CON = control total mixed ration; SEL = total mixed ration formulated with 2.5 mg Se kg^−1^ DM and 1000 IU vitamin E; SUN = total mixed ration formulated with 3% sunflower oil; SEL = total mixed ration formulated with 3% sunflower oil, 2.5 mg Se kg^−1^ DM and 1000 IU vitamin E. ^2^ Pooled standard error (SEM).

**Table 6 animals-12-01968-t006:** Serum metabolites of lactating Jersey cows fed supplements containing selenium, vitamin E and sunflower oil.

Item	Treatments ^1^				
	CON	SEL	SUN	SEL + SUN	SEM ^2^
Cholesterol, mg/dL	111	90.3	106.3	94.8	2.2
HDL ^3^, mg/dL	70.3	75.5	76.5	71.2	1.48
LDL ^4^, mg/dL	36.2 ^a^	19.0 ^c^	24.2 ^b^	22.0 ^c^	0.40
VLDL ^5^, mg/dL	4.6 ^a^	3.0 ^b^	2.6 ^c^	3.4 ^b^	0.18
TGL ^6^, mg/dL	14.4	16.1	13.5	15.3	0.27
Glucose, mg/dL	41.8	48.0	45.8	37.6	1.28

^a–c^ Means within a row with different superscripts differ by Bonferroni-test (*p* < 0.05). ^1^ Cows were provided four experimental diets: CON = control total mixed ration; SEL = total mixed ration formulated with 2.5 mg Se kg^−1^ DM and 1000 IU vitamin E; SUN = total mixed ration formulated with 3% sunflower oil; SEL = total mixed ration formulated with 3% sunflower oil, 2.5 mg Se kg^−1^ DM and 1000 IU vitamin E. ^2^ Pooled standard error (SEM).^3^ HDL = high-density lipoprotein cholesterol; ^4^ LDL = low-density lipoprotein cholesterol; ^5^ VLDL = very low-density lipoprotein cholesterol; ^6^ TGL = triglycerides.

## Data Availability

The data that support the findings of this study are available from the corresponding author, upon reasonable request.

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
