# Peer review of "Inclusion of Sunflower Oil, Organic Selenium, and Vitamin E on Milk Production and Composition, and Blood Parameters of Lactating Cows"

_animals, 2022, doi:10.3390/ani12151968_

Round 1
Reviewer 1 Report
Dear authors, this manuscript seems to be interesting, using additive to alter the nutritional value of the milk to obtain a “better” milk, however, the language and the format still need to be improved, some lines are difficult to read.
1. L 50, what is “their proportion”?
2. L50-52, why is “thus”? How can you know the effect of sunflower oil by the above information?
3. L55, what’s the meaning of “peak of lactation”
4. L56, provides
5. L56-58, what’s the meaning of “better atherogenicity and thrombogenicity index”
6. L59-60, serum levels of these antioxidants in milk?
7. As descripted in the introduction section, the treatment effect was all proven before, why this experiment was necessary?
8. The hypothesis is not in consistent with the introduction
9. L81, 646 m
10. L82, average maximun temperature?
11. L83, More details are need about the cows, like the parities, the body weight, the milk production…
12. L85-86, was sunflower oil supplemented in CON group?
13. L86, basal ration?
14. L89, was treatment 4) the combination of 2) and 3)?
15. L97, food or feed?
16. L98, what’s the meaning of “bromatological analysis”?
17. L99, was the samples pooled by cow? Or by treatments?
18. L103, what’s the difference between aNDF and NDF?
19. Were the sunflower oil, Vit E and Se add by top dressed? Or mixed in TMR?
20. Table 1., Why the CP content differed between CON and SEL group? (18.1 vs 18.6% could be a big difference)
21. Table 1., the NEL need to be provided.
22. L123, the milk sample was pooled by what? The milk yield? Or a fixed ratio?
23. L125, was the milk samples collected before the experiment? How could you know the baseline?
24. L134, 1 g, why did you use the weight, not the volume? Is the milk whey solid or liquid? How did you get the milk whey?
25. L143, what’s the meaning of “LV”
26. L148-152, rewrite
27. L156-158, what temperature was used in the analysis
28. L168, ×, not x
29. L171, -80℃
30. L173, please describe the determination of Vit E in more details
31. L178-179, rewrite please
32. Statistical analysis, what was the random effect in the model? Was the different sampling day considered as repeated measurement? Or the sampling day was also used as a fixed effect?
33. L193-197, as no P value was showed in this section, how can you describe “greater” or “lower”? only due to the differen amount?
34. L199, Lesser? Or Less?
35. L214, what effect?
36. The results and discussion section need to be improved, especially the language and format, for example, in L351-352, the HDL, VLDL and LDL used a lot befine, why did the authors define them again here?
37. L379, what performance?
38. The format of the reference section needed to be double checked.
Author Response
Dear authors, this manuscript seems to be interesting, using additive to alter the nutritional value of the milk to obtain a “better” milk, however, the language and the format still need to be improved, some lines are difficult to read.
- L 50, what is “their proportion”?
AU: Thank you. We rewrote the sentence.
- L50-52, why is “thus”? How can you know the effect of sunflower oil by the above information?
AU: We improved the sentence.
- L55, what’s the meaning of “peak of lactation”
AU: We improved the sentence. We meant peak milk yield.
- L56, provides
AU: We fixed it.
- L56-58, what’s the meaning of “better atherogenicity and thrombogenicity index”
AU: Thank you. We meant “lower atherogenicity and thrombogenicity index". We fixed it.
- L59-60, serum levels of these antioxidants in milk?
AU: We fixed it.
- As descripted in the introduction section, the treatment effect was all proven before, why this experiment was necessary?
AU: Individually, the effect of dietary sunflower oil and Se plus vitamin E are well described in the literature. However, their effects together have not been reported in dairy cows’ studies.
- The hypothesis is not in consistent with the introduction
AU: Thank you. We improved the sentence to make it clearer.
- L81, 646 m
AU: We fixed it.
- L82, average maximun temperature?
AU: We fixed it.
- L83, More details are need about the cows, like the parities, the body weight, the milk production…
AU: We added all details required.
- L85-86, was sunflower oil supplemented in CON group?
AU: We fixed it. The CON group did not receive sunflower oil and Se plus vitamin E supplementation. Thank you.
- L86, basal ration?
AU: Thank you. We fixed it.
- L89, was treatment 4) the combination of 2) and 3)?
AU: Yes, it was. We added information to make the sentence clearer.
- L97, food or feed?
AU: We fixed it. Thank you.
- L98, what’s the meaning of “bromatological analysis”?
AU: We replace it with “chemical analysis” to make it clearer.
- L99, was the samples pooled by cow? Or by treatments?
AU: The samples were pooled by cow. We added this information in the sentence.
- L103, what’s the difference between aNDF and NDF?
AU: aNDF means the NDF feed analysis was performed using amylase enzyme, which eliminates all starch contamination in the feed. Neutral Detergent Fiber (NDF) is the feed treated only by using NDF solution, without any amylase enzyme addition. In the 1990’s, alpha-amylase enzyme was added to the ND solution to further clean up the residue and give a more accurate representation of fiber in the sample.
- Were the sunflower oil, Vit E and Se add by top dressed? Or mixed in TMR?
AU: They were mixed in TMR. We added this information to the manuscript.
- Table 1., Why the CP content differed between CON and SEL group? (18.1 vs 18.6% could be a big difference)
AU: This difference may be through the feedstuff variation and due to mixing procedure variability among the treatments. Even by providing the same amounts of feedstuff to the animals, according to basal diet proportion, we detected these CP values. We believe this 0.5% percentage point difference did not impact the results herein presented, once the CP level did not play as a limiting factor for ruminal functioning.
- Table 1., the NELneed to be provided.
AU: Thank you for the suggestion. We added the NEL in table 1.
- L123, the milk sample was pooled by what? The milk yield? Or a fixed ratio?
AU: The milk samples were pooled by cow keeping a fixed ratio. We added this information in the sentence.
- L125, was the milk samples collected before the experiment? How could you know the baseline?
AU: The information that was collected and could be used as covariate parameters were: milk yield, body weight, and dry matter intake. They were collected during the adaptation period. However, we just included the DMI adjustment in the model. This information of covariate adjustment was included in the statistical analysis section. Thank you.
- L134, 1 g, why did you use the weight, not the volume? Is the milk whey solid or liquid? How did you get the milk whey?
AU: We have run the vitamin E analysis according to Liu et al. (1996). The authors determine weight rather than volume.
- L143, what’s the meaning of “LV”
AU: It was Level of Vitamin. However, the editor requested the deletion of this sentence, summarizing the methodology.
- L148-152, rewrite
AU: We rewrote the sentence as requested.
- L156-158, what temperature was used in the analysis
AU: Thank you. The temperatures used were described in the manuscript.
- L168, ×, not x
AU: We fixed it.
- L171, -80℃
AU: We fixed it.
- L173, please describe the determination of Vit E in more details
AU: Thank you for the suggestion; however, the editor requested to summarize the method as briefly as possible, only citing the author for more details.
- L178-179, rewrite please
AU: We rewrote the sentence as requested.
- Statistical analysis, what was the random effect in the model? Was the different sampling day considered as repeated measurement? Or the sampling day was also used as a fixed effect?
AU: The random effect was animal within treatment. The different sampling day was not considered as repeated measurements. It is because we performed all analyses in a pooled sample, with no time factor.
- L193-197, as no P value was showed in this section, how can you describe “greater” or “lower”? only due to the differen amount?
AU: This sentence only summarizes the descriptive chemical analysis of the experimental diets. We added the term “numerically” and “descriptive” in the sentence to make it clearer.
- L199, Lesser? Or Less?
AU: Thank you. We fixed it.
- L214, what effect?
AU: We improved the sentence.
- The results and discussion section need to be improved, especially the language and format, for example, in L351-352, the HDL, VLDL and LDL used a lot befine, why did the authors define them again here?
AU: We improved the Results and Discussion section altering the language and format as requested. Thank you.
- L379, what performance?
Au: DMI, Milk yield, and fat-corrected (3.5%) milk yield. We added this information in the sentence.
- The format of the reference section needed to be double-checked.
AU: We double-checked the reference section and fixed all the inconsistencies.

Reviewer 2 Report
Dear Editor and Authors,
I send you my review about the article “Inclusion of sunflower oil, organic selenium, and vitamin E on milk production and composition, and blood parameters of lactating cows”.
The scope of the paper, as reported in the aim was to study the effects of the inclusion of sunflower oil added with organic selenium and vitamin E in diets of lactating cows.
In my opinion, the article, although it result very well written and it is very well structured, it shows, also, a serious lack in originality.
Therefore, my opinion is that this article result suitable for publication after some revisions that I a report below.
The introduction result too much short and not adequately to the aim of the research. Thus, in the introduction should be better explained the originality of this paper. To improve the originality of the paper, I suggest to the Authors, to reports more number of article that have study the use of sunflower oil and antioxidant like selenium and vitamin E.
Moreover, after, they should stress, always in the introduction, the difference among their study and the others previously reported.
The paragraph of materials and methods result complete and well done, nevertheless, the Authors should explain why the have chosen to use as fixed factor only the effect of treatment.
Furthermore, to facilitate comprehension by the readers of the contents, in the section of statistical analysis the model used to estimate the least square means values should be reported.
Moreover, for the future, in the ANOVA post hoc analysis, I suggest to the Authors to use the method of Bonferroni or the one of Sidak to test the difference among more of two thesis.
The results is well presented, nevertheless, to improve the originality of the article the comparison to the data reported in the literature should be improved.
To this aim I suggest to the Authors to increase the number of article used in the discussion and compare data reported by they with the one reported in this study.
Moreover, the Authors, should highlight the new acquisitions of their research with respect to the article mentioned.
Finally, always to facilitate a better understanding of the originality of this article should be highlight also in the conclusions the new information gained in this research respect to the oldest ones
Best regards
Author Response
Dear Editor and Authors,
I send you my review about the article “Inclusion of sunflower oil, organic selenium, and vitamin E on milk production and composition, and blood parameters of lactating cows”.
The scope of the paper, as reported in the aim was to study the effects of the inclusion of sunflower oil added with organic selenium and vitamin E in diets of lactating cows.
In my opinion, the article, although it result very well written and it is very well structured, it shows, also, a serious lack in originality.
AU: We highlighted the originality of this study in the introduction, Results and Discussion, and Conclusion sections. Thank you.
Therefore, my opinion is that this article result suitable for publication after some revisions that I a report below.
The introduction result too much short and not adequately to the aim of the research. Thus, in the introduction should be better explained the originality of this paper. To improve the originality of the paper, I suggest to the Authors, to reports more number of article that have study the use of sunflower oil and antioxidant like selenium and vitamin E.
Moreover, after, they should stress, always in the introduction, the difference among their study and the others previously reported.
AU: Thank you for the suggestion. We modified the introduction structure to improve its readability. To date, only one paper describes the sunflower oil and Selenium plus Vitamin supplementation for lactating dairy cows; however, it does not present performance results of the lactating dairy cows. Thus, this study demonstrates high originality in the literature through the impact of sunflower oil plus antioxidants supplementation on the performance of dairy cows.
The paragraph of materials and methods result complete and well done, nevertheless, the Authors should explain why they have chosen to use as fixed factor only the effect of treatment.
AU: The initial information that was collected and could be used as covariate parameters were: milk yield, body weight, and dry matter intake. They were collected during the adaptation period. However, we just included the DMI adjustment in the model. This information on covariate adjustment was included in the statistical analysis section. Thank you. All other effects such as parity, body weight, and milk yield did not work as covariate adjustment factors.
Furthermore, to facilitate comprehension by the readers of the contents, in the section of statistical analysis the model used to estimate the least square means values should be reported.
AU: We added the statistical model as requested. Thank you.
Moreover, for the future, in the ANOVA post hoc analysis, I suggest to the Authors to use the method of Bonferroni or the one of Sidak to test the difference among more of two thesis.
AU: We performed the Bonferroni test as suggested. However, the P-values changed but the results did not change.
The results is well presented, nevertheless, to improve the originality of the article the comparison to the data reported in the literature should be improved. To this aim I suggest to the Authors to increase the number of article used in the discussion and compare data reported by they with the one reported in this study.
AU: To date, only one paper reported the effect of sunflower oil and Se plus vitamin E supplementation in dairy cows. So, we discuss n biological effect of each supplement (sunflower oil, Se, and vitamin E) by comparing some studies that investigated their individual effect.
Moreover, the Authors, should highlight the new acquisitions of their research with respect to the article mentioned.
AU: We highlighted the acquisitions of this study in the Results and Discussion section. Also, we added a sentence in the introduction section clearly demonstrating our aim with this study.
Finally, always to facilitate a better understanding of the originality of this article should be highlight also in the conclusions the new information gained in this research respect to the oldest ones
AU: We added a sentence in the Conclusion section highlighting the new information gained with this study. Thank you for all your suggestions.
